# Concept Distillation: Leveraging Human-Centered Explanations for Model Improvement

**Avani Gupta**[1,2,†]   **Saurabh Saini**[2]   **P J Narayanan** [2]
[1]M42, UAE    [2]IIIT Hyderabad, India
{avani.gupta, saurabh.saini}@research.iiit.ac.in
pjn@iiit.ac.in

## Abstract

Humans use abstract *concepts* for understanding instead of hard features. Recent interpretability research has focused on human-centered concept explanations of neural networks. Concept Activation Vectors (CAVs) estimate a model's sensitivity and possible biases to a given concept. In this paper, we extend CAVs from post-hoc analysis to ante-hoc training in order to reduce model bias through fine-tuning using an additional *Concept Loss*. Concepts were defined on the final layer of the network in the past. We generalize it to intermediate layers using class prototypes. This facilitates class learing in the last convolution layer which is known to be most informative. We also introduce *Concept Distillation* t create richer concepts using a pre-trained knowledgeable model as the teacher. Our method can sensitize or desensitize a model towards concepts. We show applications of concept-sensitive training to debias several classification problems. We also use concepts to induce prior knowledge into IID, a reconstruction problem. Concept-sensitive training can improve model interpretability, reduce biases, and induce prior knowledge. Please visit https://avani17101.github.io/Concept-Distilllation/ for code and more details.

## 1 Introduction

EXplainable Artificial Intelligence (XAI) methods are useful to understand a trained model's behavior [74]. They open the black box of Deep Neural Networks (DNNs) to enable post-training identification of unintended correlations or biases using similarity scores or saliency maps. Humans, however, think in terms of abstract *concepts*, defined as groupings of similar entities [27]. Recent efforts in XAI have focused on concept-based model explanations to make them more aligned with human cognition. Kim et al. [32] introduce Concept Activation Vectors (CAVs) using a concept classifier hyperplane to quantify the importance given by the model to a particular concept. For instance, CAVs can determine the model's sensitivity on 'striped-ness' or 'dotted-ness' to classify Zebra or Cheetah using user-provided concept samples. They measure the concept sensitivity of the model's final layer prediction with respect to intermediate layer activations (outputs). Such post-hoc analysis can evaluate the transparency, accountability, and reliability of a learned model [14] and can identify biases or unintended correlations acquired by the models via shortcut learning [32, 5].

The question we ask in this paper is: If CAVs can identify and quantify sensitivity to concepts, can they also be used to improve the model? Can we learn less biased and more human-centered models? In this paper, we extend CAVs to ante-hoc model improvement through a novel *concept loss* to desensitize/sensitize against concepts. We also leverage the broader conceptual knowledge of a large pre-trained model as a teacher in a *concept distillation* framework for it.

---

[†]Work done while at IIIT Hyderabad

37th Conference on Neural Information Processing Systems (NeurIPS 2023).

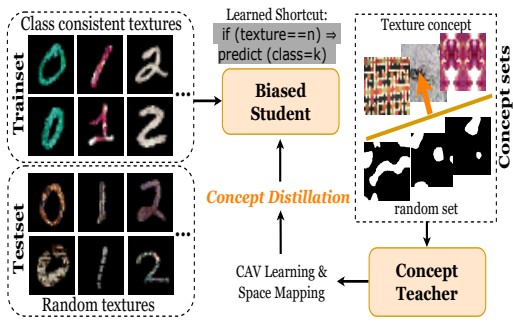

Figure 1: Overview of our approach: The generic conceptual knowledge of a capable teacher can be distilled to a student for performance improvement through bias removal and prior induction.

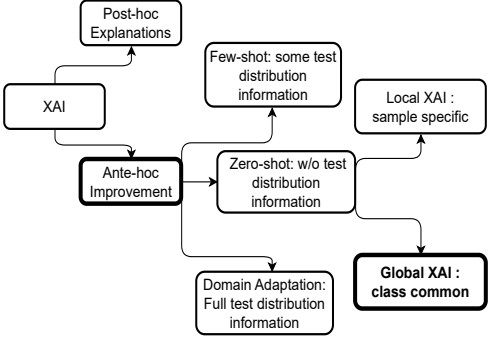

Figure 2: Categorization of related works with ours highlighted.

XAI has been used for ante-hoc model improvement during training [35, 30, 4]. They typically make fundamental changes to the model architecture or need significant concept supervision, making extensions to other applications difficult. For example, Koh et al. [35] condition the model by first predicting the underlying concept and then using it for class prediction. Our method can sensitize or desensitize the model to user-defined concepts without modifications to the architecture or direct supervision.

Our approach relies on the sensitivity of a trained model to human-specified concepts. We want the model to be sensitive to relevant concepts and indifferent to others. For instance, a cow classifier might be focusing excessively on the grass associated with cow images. If we can estimate the sensitivities of the classifier to different concepts, we can steer it away from irrelevant concepts. We do that using a *concept loss* term $L_C$ and fine-tuning the trained base model with it. Since the base models could be small and biased in different ways, we use *concept distillation* using a large, pre-trained teacher model that understands common concepts better.

We also extend concepts to work effectively on intermediate layers of the model, where the sensitivity is more pronounced. Kim et al. [32] measure the final layer's sensitivity to any intermediate layer outputs. They ask the question: if any changes in activations are done in the intermediate layer, what is its effect on the final layer prediction? They used the final layer's loss/logit to estimate the model sensitivity as their interest was to study concept sensitivities for interpretable model prediction. We, on the other hand, aim to fine-tune a model by (de)sensitizing it towards a given concept which may be strongest in another layer [3]. Thus, it is crucial for us to measure the sensitivity in *any* layer by evaluating the effect of the changes in activations in one intermediate layer on another. We employ prototypes or average class representations in that layer for this purpose. Prototypes are estimated by clustering the class sample activations [12, 72, 41, 49, 48, 63, 43]. Our method, thus, allows intervention in any layer.

In this paper, we present a simple but powerful framework for model improvement using concept loss and concept distillation for a user-given concept defined in any layer of the network. We leverage ideas from post-hoc global explanation techniques and use them in an ante-hoc setting by encoding concepts as CAVs via a teacher model. Our method also admits sample-specific explanations via a local loss [52] along with the global concepts whenever possible. We improve state-of-the-art on classification problems like ColorMNIST and DecoyMNIST [51, 16, 52, 5], resulting in improved accuracies and generalization. We introduce and benchmark on a new and more challenging TextureMNIST dataset with texture bias associated with digits. We demonstrate concept distillation on two applications: *(i)* debiasing extreme biases on classification problems involving synthetic MNIST datasets [42, 16] and complex and sensitive age-*vs.*-gender bias in the real-world gender classification on BFFHQ dataset [34] and *(ii)* prior induction by infusing domain knowledge in the reconstruction problem of Intrinsic Image Decomposition (IID) [38] by measuring and improving disentanglement of albedo and shading concepts. To summarize, we:

- Extend CAVs from post-hoc explanations to ante-hoc model improvement method to sensitize/desensitize models on specific concepts without changing the base architecture.

- Extend the model CAV sensitivity calculation from only final layer to *any* layer and enhance it by making it more global using prototypes.
- Introduce concept distillation to exploit the inherent knowledge of large pretrained models as a teacher in concept definition.
- Benchmark results on standard biased MNIST datasets and on a challenging TextureMNIST dataset that we introduce.
- Show application on a severely biased classification problem involving age bias.
- Show application beyond classification to the challenging multi-branch Intrinsic Image Decomposition problem by inducing human-centered concepts as priors. To the best of our knowledge, this is the first foray of concept-based techniques into non-classification problems.

## 2   Related Work

The related literature is categorized in Fig. 2. Post-hoc (after training) explainability methods include Activation Maps Visualization [59], Saliency Estimation [54], Model Simplification [69], Model Perturbation [17], Adversarial Exemplar Analysis [22], *etc*. See recent surveys for a comprehensive discussion [74, 15, 45, 65, 2]. Different concept-based interpretability methods are surveyed by Hitzler and Sarker [27], Schwalbe [58], Holmberg et al. [28].

The ante-hoc model improvement techniques can be divided into zero-shot [70], few-shot [67], and multi-shot (domain-adaptation [66]) categories based on the amount of test distribution needed. In zero-shot training, the model never sees the distribution of test samples [70], while in few-shot, the model has access to some examples from the distribution of the test set [67]. Our method works with abstract concept sets (different than any test sample), being essentially a zero-shot method but can take advantage of few-shot examples, if available, as shown in results on BFFHQ subsection 4.1. We restrict the discussion to the most relevant model improvement methods.

Ross et al. [52] penalize a model if it does not prefer the Right answers for Right Reasons (RRR) by using explanations as constraints. EG Erion et al. [16] augment gradients with a penalizing term to match with user-provided binary annotations. More details are available in relevant surveys [18, 68, 24, 8].

Based on the scope of explanation, the existing XAI methods can be divided into global and local methods [74]. Global methods [32, 20] provide explanations that are true for all samples of a class (*e.g*. 'stripiness' concept is focused for the zebra class or 'red color' is focused on the prediction of zero) while local methods [1, 59, 60, 53] explain each of samples individually often indicating regions or patches in the image which lead the model to prediction (example, this particular patch (red) in this image was focused the most for prediction of zero). Kim et al. [32] quantify concept sensitivity using CAVs followed by subsequent works [62, 73, 57]. Kim et al. [32] sample sensitivity is local (sample specific) while they aggregate class samples for class sample sensitivity estimation, which makes their method global (across classes). We enhance their local sample sensitivity to the global level via using prototypes that capture class-wide characteristics by definition [12].

Only a few concept-oriented deep learning methods train neural networks with the help of concepts [56, 55]. Concept Bottleneck Models (CBMs) [35], Concept-based Model Extraction (CME) [30], and Self-Explaining Neural Networks (SENNs) [4] predict concepts from inputs and use them to infer the output. CBMs [35] require concept supervision, but CME [30] can train in a partially supervised manner to extract concepts combining multiple layers. SENN [4] learns interpretable basis concepts by approximating a model with a linear classifier. These methods require architectural modifications of the base model to predict concepts. Their primary goal is interpretability, which differs from our model improvement goal using specific concepts *without* architectural changes. Closest to our work is ClArC [5], which leverages CAVs to manipulate model activations using a linear transformation to remove artifacts from the final results. Unlike them, our method trains the DNN to be sensitive to *any* specific concept, focusing on improving generalizability for multiple applications.

DFA [40] and EnD [64] use bias-conflicting or out-of-distribution (OOD) samples for debiasing using few-shot generic debiasing. DFA uses a small set of adversarial samples and separate encoders for learning disentangled features for intrinsic and biased attributes. EnD uses a regularization strategy to prevent the learning of unwanted biases by inserting an information bottleneck using pre-known bias types. It is to be noted that both DFA and EnD are neither zero-shot nor interpretability-based methods. In comparison, our method works in both known bias settings using abstract user-provided

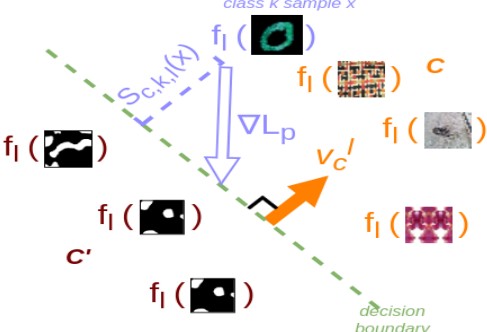

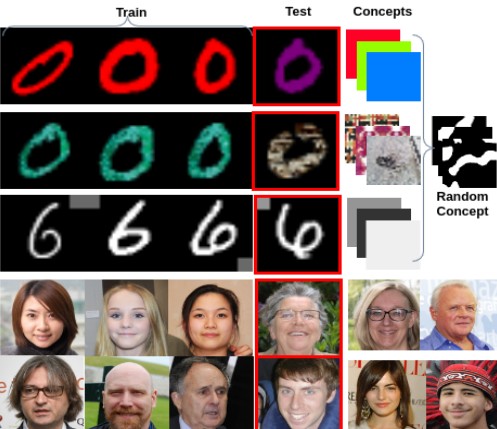

Figure 3: $v_c^l$ is calculated as normal to the separating hyperplane of concept set activations (textures $C$ *vs.* random set $C'$ here). A model biased towards $C$ will have its class samples's loss gradient $\nabla L_p$ along $v_c^l$ (measured by sensitivity $S_{C,k,l}(x)$). To desensitize the model for $C$, we perturb $\nabla L_p$ to be parallel to the decision boundary by minimizing the cosine of the projection angle.

Figure 4: Datasets used: ColorMNIST (top row), TextureMNIST (next row), DecoyMNIST (third row), and BFFHQ (bottom rows). Concepts used include color, textured and gray patches, and bias-conflicting samples shown on the right.

concept sets and unknown bias settings using the bias-conflicting samples. Our Concept Distillation method is a *concept sensitive training* method for induction of concepts into the model, with debiasing being an application. Intrinsic Image Decomposition (IID) [11, 19] involves decomposing an image into its constituent Reflectance ($R$) and Shading ($S$) components [38, 46] which are supposed to be disentangled [46, 23].

We use CAVs for inducing R and S priors in a pre-trained SOTA IID framework [44] with improved performance. To the best of our knowledge, we are the first to introduce prototypes for CAV sensitivity enhancement. However, prototypes have been used in the interpretability literature before to capture class-level characteristics [33, 71, 31] and also have been used as pseudo-class labels before [12, 72, 41, 49, 48, 63, 43]. Some recent approaches [29, 61] use generative models to generate bias-conflicting samples (*e.g.* other colored zeros in ColorMNIST) and train the model on them to remove bias. Specifically, Jain et al. [29] use SVMs to find directions of bias in a shared image and language space of CLIP [50] and use Dall-E on discovered keywords of bias to generate bias-conflicting samples and train the model on them. Song et al. [61] use StyleGAN to generate bias-conflicting samples. Training on bias-conflicting samples might not always be feasible due to higher annotation and computation costs.

One significant difference between such methods [29, 61] is that they propose data augmentation as a debiasing strategy, whereas we directly manipulate the gradient vectors, which is more interpretable. Moayeri et al. [47] map the activation spaces of two models using the CLIP latent space similarity. Due to the generality of the CLIP latent space, this approach is helpful to encode certain concepts like 'cat, dog, man,' but it is not clear how it will work on abstract concepts with ambiguous definitions like 'shading' and 'reflectance' as seen in the IID problem described above.

## 3 Concept Guidance and Concept Distillation

Concepts have been used to explain model behavior in a post-hoc manner in the past. Response to abstract concepts can also demonstrate the model's intrinsic preferences, biases, etc. Can we use concepts to guide the behavior of a trained base model in desirable ways in an ante-hoc manner? We describe a method to add a *concept loss* to achieve this. We also present concept distillation as a way to take advantage of large foundational models with more exposure to a wide variety of images.

### 3.1 Concept Sensitivity to Concept Loss

Building on Kim et al. [32], we represent a concept $C$ using a Concept Activation Vector (CAV) as the normal $v_C^l$ to a linear decision boundary between concept samples $C$ from others $C'$ in a layer $l$ of the model's activation space (Fig. 3). The model's sensitivity $S_{C,l}(\boldsymbol{x}) = \nabla L_o\left(f_l(\boldsymbol{x})\right) \cdot v_C^l$ to $C$ is

the directional derivative of final layer loss $L_o$ for samples $\boldsymbol{x}$ along $v_C^l$ [32]. The sensitivity score quantifies the concept's influence on the model's prediction. A high sensitivity for color concept may indicate a color bias in the model.

These scores were used for post-hoc analysis before ([32]). We use them ante-hoc to desensitize or sensitize the base model to concepts by perturbing it away from or towards the CAV direction (Fig. 3). The gradient of loss indicates the direction of maximum change. Nudging the gradients away from the CAV direction encourages the model to be less sensitive to the concept and vice versa. For this, we define a concept loss $L_C$ as the absolute cosine of the angle between the loss gradient and the CAV direction

$$L_C(\boldsymbol{x}) = |\cos(\nabla L_o\left(f_l(\boldsymbol{x})\right), \boldsymbol{v}_C^l)|, \tag{1}$$

which is minimized when the CAV lies on the classifier hyperplane (Fig. 3). We use the absolute value to not introduce the opposite bias by pushing the loss gradient in the opposite direction. A loss of $(1 - L_C(x))$ will sensitize the model to $C$. We fine-tune the trained base model for a few epochs using a total loss of $L = L_o + \lambda L_C$ to desensitize it to concept $C$, where $L_o$ the base model loss.

### 3.2 Concepts using Prototypes

Concepts can be present in any layer $l$, though the above discussion focuses on the sensitivity calculation of the final layer using model loss $L_o$. The final convolutional layer is proven to learn concepts better than other layers [3]. We can estimate the concept sensitivity of any layer using a loss for that layer. How do we get a loss for an intermediate layer, as no ground truth is available for it?

Class prototypes have been used as pseudo-labels in intermediate layers before [12, 72, 41]. We adapt prototypes to define a loss in intermediate layers. Let $f_l(x)$ be the activation of layer $l$ for sample $x$. We group the $f_l(x)$ values of the samples from each class into K clusters. The cluster centers $P_i$ together form the prototype for that class. We then define prototype loss for each training sample $x$ using the prototype corresponding to its class as

$$L_p(x) = \frac{1}{K} \sum_{k=1}^{K} \|f_l(x) - P_k\|^2. \tag{2}$$

We use $L_p$ in place of $L_o$ in Eq. 1 to define the concept loss in layer $l$. The prototype loss facilitates the use of intermediate layers for concept (de)sensitization. Experiments reported in Tab. 1 confirm the effectiveness of $L_p$. Prototypes also capture sample sensitivity at a global level using all samples of a class beyond the sample-specific levels. We update the prototypes after a few iterations as the activation space evolves. If $P^n$ is the prototype at Step $n$ and $P^c$ the cluster centres using the current $f_l(x)$ values, the next prototype is $P_k^{n+1} = (1 - \alpha)P_k^n + \alpha P_k^c$ for each cluster $k$.

### 3.3 Concept Distillation using a teacher

Concepts are learned from the base model in the above formulation. Base models may have wrong concept associations due to their training bias or limited exposure to concepts. Can we alleviate this problem using a larger model that has seen vast amounts of data as a teacher in a distillation framework?

We use the DINO [13], a self-supervised model trained on a large number of images, as the teacher and the base model as the student for concept distillation. The teacher and student models typically have different activation spaces. We map the teacher space to the student space before concept learning. The mapping uses an autoencoder [26] consisting of an encoder $E_M$ and a decoder $D_M$ (Fig. 5). As a first step, the autoencoder (Fig. 5) is trained to minimize the loss $L_{D_M} + L_{E_M}$. $L_{D_M}$ is the pixel-wise L2 loss between the original ($f_t$) and decoded ($\hat{f}_t$) teacher activations and $L_{E_M}$ is the pixel-wise L2 loss between the mapped teacher ($\hat{f}_s$) and the student ($f_s$) activations. The mapping is learned over the concept set of images $C$ and $C'$. See the dashed purple lines in Fig. 5.

Next, we learn the CAVs in the distilled teacher space $\hat{f}_s$, keeping the teacher, student, and mapping modules fixed. This is computationally light as only a few (50-150) concept set images are involved. The learned CAV is used in concept loss given in Eq. 1. Please note that $E_M$ is used only to align the two spaces and can be a small capacity encoder, even a single layer trained in a few epochs.

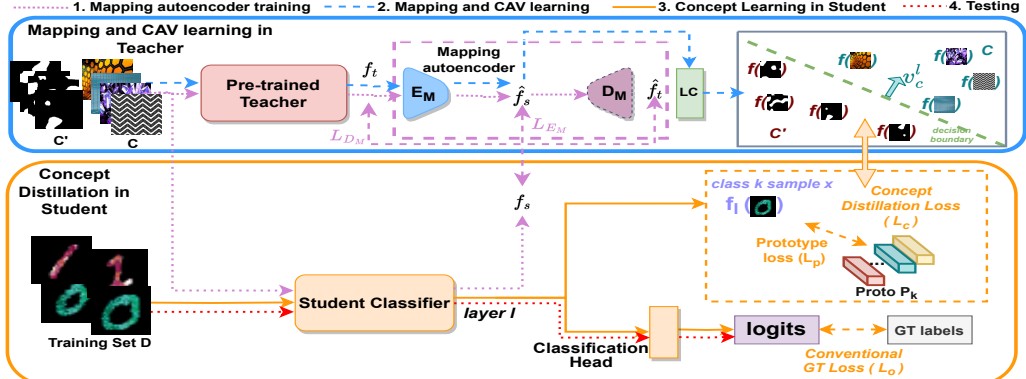

Figure 5: Our framework comprises a concept teacher and a student classifier and has the following four steps: 1) Mapping teacher space to student space for concepts $C$ and $C'$ by training an autoencoder $E_M$ and $D_M$ (dotted purple lines); 2) CAV ($\boldsymbol{v}_c^l$) learning in mapped teacher space via a linear classifier LC (dashed blue lines); 3) Training the student model with Concept Distillation (solid orange lines): We use $\boldsymbol{v}_c^l$ and class prototypes loss $L_p$ to define our concept distillation loss $L_c$ and use it with the original training loss $L_o$ to (de)sensitize the model for concept $C$; 4) Testing where the trained model is applied (dotted red lines)

---

**Algorithm 1 Concept Distillation Pipeline**

---

**Given:** A pretrained student model which is to be fine-tuned for concept (de)sensitization and a pretrained teacher model which will be used for concept distillation. Known Concepts $C$ and negative counterparts (or random samples) $C'$, student training dataset $\mathcal{D}$, and student bottleneck layer $l$, #iterations to recalculate CAVs cav_update_frequency, #iterations to update prototypes proto_update_frequency.

  1: **Concept Distillation**:
  2:      For all class samples in $\mathcal{D}$, estimate class prototypes $P_{k\in\{0,K\}}^0$ with K-means.
  3:      Current iteration $n = 0$, initial prototypes $P^0 = P^c$.
  4:    **While** not converge **do**:
  5:      **If** n = 0 or (update_cavs and n % cav_update_frequency = 0) **then**:
  6:        **Learn Mapping module**:
  7:          Forward pass $x \in C \cup C'$ from Teacher and Student to get their concept activations $f_t$ and $f_s$.
  8:          Learn the mapping module as autoencoders $E_M$ and $D_M$.
  9:        **CAV learning in mapped teacher's space**:
10:          $\mathbf{v}_C^l$ learned by binary linear classifier as normal to decision boundary of $E_M(f_t(x))$ for $x \in C$ vs $E_M(f_t(x'))$ for $x' \in C'$.
11:      **If** $n$ % proto_update_frequency $= 0$ and $n \neq 0$ **then**:
12:        Estimate new class prototypes $P_{k\in\{0,K\}}^c$ with K-means.
13:        Weighted Proto-type $P_k^{n+1} = (1-\alpha)P_k^n + \alpha P_k^c$.
14:      **Else**:
15:        $P_k^{n+1} = P_k^n$
16:      Train student with loss $L_C + L_o$.
17:      $n+ = 1$.

---

## 4 Experiments

We demonstrate the impact of the concept distillation method on the debiasing of classification problems as well as improving the results of a real-world reconstruction problem. Classification experiments cover two categories of Fig. 2: the zero-shot scenario, where no unbiased data is seen, and the few-shot scenario, which sees a few unbiased data samples. We use pre-trained DINO ViT-B8 transformer [13]* as the teacher. We use the last convolution layer of the student model for concept distillation. The mapping module ($< 120K$ parameters, $< 10$MB), CAV estimations (logistic regression, $< 1$MB), and prototype calculations all complete within a few iterations of training, taking 15-30 secs on a single 12GB Nvidia 1080 Ti GPU*. Our framework is computationally light. In our experimentation reported below, we fix the CAVs as initial CAVs and observe similar results as

---

*More details in supplementary

Table 1: Main components of our method shown on ColorMNIST dataset: Ours usage of teacher and proto-types yields the best performance.

| Teacher? | Prototype? | Accuracy |
|----------|-----------|----------|
| ✗ | ✗ | 9.96 |
| ✗ | ✓ | 26.97 |
| ✓ | ✗ | 30.94 |
| ✓ | ✓ | **50.93** |

Table 2: TCAV scores of bias concept: Our concept sensitive training significantly decreases the sensitivity of model towards bias.

| Dataset | Concept | Base Model | Ours |
|---------|---------|-----------|------|
| ColorMNIST | Color | 0.52 | **0.21** |
| DecoyMNIST | Spatial patches | 0.57 | **0.45** |
| TextureMNIST | Textures | 0.68 | **0.43** |
| BFFHQ | Age | 0.78 | **0.13** |

varying them. Also, our concept sensitive fine-tuning method converges quickly, and hence updating CAVs in every few iterations does not help[*]. Where relevant, we report average accuracy over five training runs with random seeds ($\pm$ indicates variance).

## 4.1 Concept Sensitive Debiasings

We show results on two standard biased datasets (ColorMNIST [42] and DecoyMNIST [16]) and introduce a more challenging TextureMNIST dataset for quantitative evaluations. We also experimented on a real-world gender classification dataset BFFHQ [34] that is biased based on age. We compare with other state-of-the-art interpretable model improvement methods [51, 52, 16, 64, 40, 5]. Fig. 4 summarizes the datasets, their biases, and the concept sets used.

**Poisoned MNIST Datasets: ColorMNIST** [42] has MNIST digit classes mapped to a particular color in the training set [51]. The colors are reversed in the test set. The baseline CNN model (*Base*) trained on ColorMNIST gets 0% accuracy on the 100% poisoned test set, indicating that the model learned color shortcuts instead of digit shapes. We debias the model using a concept loss for color using color patches *vs.* random shapes (negative concept) to estimate CAV for color (Fig. 4). We show results comparison on the usage of teacher and prototypes component of our method in Tab. 1. As can be seen from Tab. 1, our method of using intermediate layer sensitivity via prototypes as described in subsection 3.2 yields better results. Similarly, usage of teacher (described in subsection 3.3) facilitates better student concept learning (also shown in CAV comparisons in supplementary). Our concept sensitive training not only improves student accuracy but observes evident reduction in TCAV scores [32] of bias concept as seen from Tab. 2.

Tab. 3 compares our results with the best zero-shot interpretability based methods, *i.e.*, CDEP [51], RRR [52], and EG [16]. Our method improves the accuracy from 31% by CDEP to 41.83%, and further to 50.93% additionally with the local explanation loss from [52][*].

Table 3: Comparison of the accuracy of our method with other zero-shot interpretable model-improvement methods. All methods require user-level intervention: Our method requires concept sets, while others (CDEP, RRR, EG) require user-provided rules.

| Dataset | Bias | Base | CDEP[51] | RRR[52] | EG[16] | Ours w/o Teacher | Ours | Ours+L |
|---------|------|------|----------|---------|--------|------------------|------|--------|
| ColorMNIST | Digit color | 0.1 | 31.0 | 0.1 | 10.0 | 26.97 | 41.83 | $\mathbf{50.93}_{\pm 1.42}$ |
| DecoyMNIST | Spatial patches | 52.84 | 97.2 | **99.0** | 97.8 | 87.49 | 98.58 | $\mathbf{98.98}_{\pm 0.20}$ |
| TextureMNIST | Digit textures | 11.23 | 10.18 | 11.35 | 10.43 | 38.72 | 48.82 | $\mathbf{56.57}_{\pm 0.79}$ |

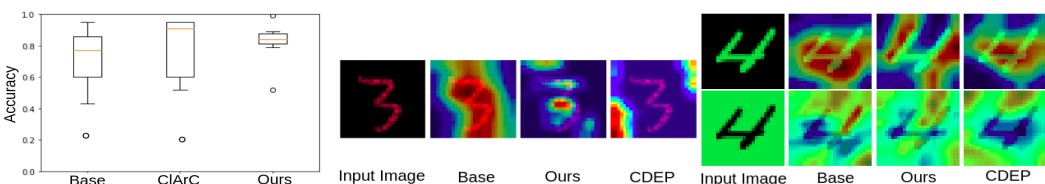

Figure 6: Left: Comparison of our method and ClArC [5] on 20% biased ColorMNIST. Right: GradCAM [59] visualizations (more red, more important) on *(i)* TextureMNIST (middle image) *(ii)* Extreme case when the color bias is in the background (bottom) instead of foreground (top). Our model focuses on the shape more than CDEP (which is more blue in the foreground).

Table 4: Our method improves the model using human-centered concepts and shows better generalization to different datasets, while CDEP, which uses pixel-wise color rules, cannot.

| | ColorMNIST Trained | | | TextureMNIST Trained | | |
|---|---|---|---|---|---|---|
| Test Dataset | Base | CDEP[51] | Ours+L | Base | CDEP[51] | Ours+L |
| Invert color | 0.00 | 23.38 | **50.93** | 11.35 | 10.18 | **45.36** |
| Random color | 16.63 | 37.40 | **46.62** | 11.35 | 10.18 | **64.96** |
| Random texture | 15.76 | 28.66 | **32.30** | 11.35 | 10.18 | **56.57** |
| Pixel-hard | 15.87 | 33.11 | **38.88** | 11.35 | 10.18 | **61.29** |

GradCAM [59] visualizations in Fig. 6 show that our trained model focuses on highly relevant regions. Fig. 6 left compares our class-wise accuracy on 20% biased ColorMNIST with ClArC [5], a few-shot global method that uses CAVs for artifact removal. (ClArC results are directly reported from their main paper due to the unavailability of the public codebase.) ClArC learns separate CAVs for each class by separating the biased color digit images (*e.g.* red zeros) from the unbiased images (*e.g.* zeros in all colors). A feature space linear transformation is then applied to move input sample activations away/towards the learned CAV direction. Their class-specific CAVs definition can not be generalized to test sets with multiple classes. This results in a higher test accuracy variance as seen in Fig. 6. Further, as they learn CAVs in the biased model's activation space, a common concept set cannot be used across classes. **DecoyMNIST** [16] has class indicative gray patches on image boundary that biased models learn instead of the shape. We define concept sets as gray patches *vs.* random set (Fig. 4) and report them in Tab. 3 second row. All methods perform comparably on this task as the introduced bias does not directly corrupt the class intrinsic attributes (shape or color), making it easy to debias. **TextureMNIST** is a more challenging dataset that we have created for further research in the area*. Being an amalgam of colors and patterns, textures are more challenging as a biasing attribute. Our method improves the performance while others struggle on this task (Tab. 3 last row). Fig. 6 shows that our method can focus on the right aspects for this task.

**Generalization** to multiple biasing situations is another important aspect. Tab. 4 shows the performance on different types of biases by differently debiased models. Our method performs better than CDEP in all situations. Interestingly, our texture-debiased model performs better on color biases than CDEP debiased on the color bias! We believe this is because textures inherently contain color information, which our method can leverage efficiently. Our method effectively addresses an extreme case of color bias introduced in the background of ColorMNIST, a dataset originally biased in foreground color. By introducing the same color bias to the background and evaluating our ColorMNIST debiased models, we test the capability of our approach. In an ideal scenario where color debiasing is accurate, the model should prioritize shape over color, a result achieved by our method but not by CDEP as shown in Fig. 6.

**BFFHQ** [34] dataset is used for the gender classification problem. It consists of images of young women and old men. The model learns entangled age attributes along with gender and gets wrong predictions on the reversed test set *i.e.* old women and young men. We use the bias conflicting samples by Lee et al. [40] – specifically, old women *vs.* women and young men *vs.* men – as class-specific concept sets.We compare against recent debiasing methods EnD [64] and DFA [40]. Tab. 5 shows our method getting a comparable accuracy of 63%. We also tried other concept set combinations: *(i)* old *vs.* young (where both young and old concepts should not affect) → 62.8% accuracy, *(ii)* old *vs.* mix (men and women of all ages) and young *vs.* mix → 62.8% accuracy, *(iii)* old *vs.* random set (consisting of random internet images from [32]) and young *vs.* random → 63% accuracy. These experiments indicate the stability of our method to the concept set definitions. This proves our method can also work with bias-conflicting samples [40] and does not necessarily require concept sets (Tab. 5). We also experimented by training class-wise (for all women removing young bias

Table 5: Comparisons in few-shot setting: Our method is not limited to user-provided concept sets and can also work with bias-conflicting samples. We compare our accuracy over the BFFHQ dataset [34] with other few-shot debiasing methods.

| Dataset | Bias | Base | EnD [64] | DFA [40] | Ours w/o Teacher | Ours |
|---|---|---|---|---|---|---|
| BFFHQ | Age | $56.87_{\pm2.69}$ | $56.87_{\pm1.42}$ | $61.27_{\pm3.26}$ | 59.4 | $\mathbf{63}_{\pm\mathbf{0.79}}$ |

Table 6: Increasing the bias of the teacher on ColorMNIST reduces accuracy.

| Teacher's Training Data Bias% | No Distil | 5 | 10 | 25 | 75 | 90 | 100 |
|---|---|---|---|---|---|---|---|
| Student Accuracy% | 26.97 | 40.47 | 33.18 | 30.38 | 28.74 | 23.23 | 23.63 |

followed by men removing old bias and vice versa) vs training for all classes together (both men, women removing age bias as described above) and observed similar results, suggesting that our concept sensitive training is robust to class-wise or all class agnostic training. Local interpretable improvement methods like CDEP, RRR, and EG are not reported here as they cannot capture complex concepts like age due to their pixel-wise loss or rule-based nature.

**Discussion** *No Distillation Case:* We additionally show our method without teacher (Our Concept loss with CAVs learned directly in student) in all experiments as "Ours w/o Teacher" and find inferior performance when compared to Our method with Distillation. **Bias in Teacher:** We check the effect of bias in teacher by training it with varying fractions of OOD samples to bias them. Specifically, we use the same student architecture for the teacher. The teacher is trained on the ColorMNIST dataset with 5, 10, 25, 50, 75, and 90% biased color samples in the trainset (*e.g.* $k\%$ bias indicates $k\%$ red zeros). The resulting concept-distilled system is then tested on the standard 100% reverse color setting of ColorMNIST. Tab. 6 shows that concept distillation improves performance even with high teacher bias, though accuracy decreases with the increasing bias in teacher. Apparently, 100% bias in teacher in this setting of teacher with same architecture as the student is essentially CAV learning in same model case (No Distil). Here, the improvements are due to prototypes, and as can be seen, there is a slight degradation in performance of *100% bias* vs *No Distil* (23.63 vs 26.97). This can be attributed to an error due to the mapping module.

## 4.2 Prior Knowledge Induction

Intrinsic Image Decomposition (IID) is an inverse rendering problem [7] based on the retinex theory [39], which suggests that an image $I$ can be divided into Reflectance $R$ and Shading $S$ components such that $I = R \cdot S$ at each pixel. Here $R$ represents the material color or albedo, and $S$ represents the scene illumination at a point. As per definition, $R$ and $S$ are *disentangled*, with $R$ invariant to illumination and $S$ invariant to albedo. As good ground truth for IID is hard to create, IID algorithms are evaluated on synthetic data or using some sparse manual annotations [9, 36]. Gupta et al. [23] use CAV-based sensitivity scores to measure the disentanglement of $R$ and $S$. They create concept sets of *(i) varying albedo*: wherein the material color of the scene is varied *(ii) varying illumination*: where illumination is varied. From the definition of IID, Reflectance shall only be affected by albedo variations and not by illumination variations and vice-versa for Shading. They defined *Concept Sensitivity Metric (CSM)* to measure $R$-$S$ disentanglement and evaluate IID methods post-hoc. Using our concept distillation framework, we extend their post-hoc quality evaluation method to ante-hoc training of the IID network to increase disentanglement between $R$ and $S$. We train in different experimental settings wherein we only train the $R$ branch ($R$ only) and both $R$ and $S$ branches together ($R\&S$) with our concept loss in addition to the original loss [44] and report results in Tab. 7.

We choose the state-of-the-art CGIID [44] network as the baseline. The last layer of both $R$ and $S$ branches is used for concept distillation, and hence no prototypes are used. Following Li and Snavely [44], we fine-tune over the CGIntrinsics [44], IIW [10], and SAW [37] datasets while we report results

Table 7: IID performance on ARAP dataset: Inducing human-centered concepts like albedo-invariance of S and illumination invariance of R results in improved IID performance.

| Model | MSE R ↓ | MSE S ↓ | SSIM R ↑ | SSIM S ↑ | Synthetic | | Real-World |
|---|---|---|---|---|---|---|---|
| | | | | | CSM R ↑ | CSM S ↑ | CSM R ↑ |
| CGIID [44] | 0.066 | **0.027** | 0.536 | 0.581 | 1.790 | 0.930 | 5.431 |
| CGIID++ | 0.080 | 0.032 | 0.520 | 0.552 | 0.860 | 0.401 | 3.421 |
| Ours (R only) | **0.052** | **0.027** | **0.54** | 0.581 | 1.889 | **1.260** | 4.89 |
| Ours (R & S) | 0.059 | 0.028 | 0.538 | **0.586** | **6.040** | 1.023 | **78.46** |

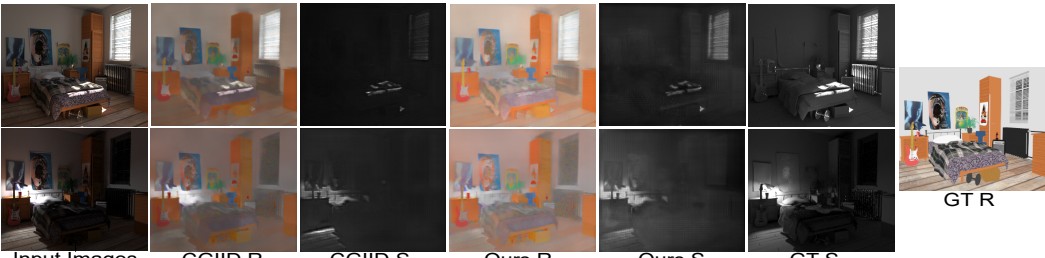

Figure 7: Qualitative IID results: Our method uses complex concepts like albedo and illumination to enhance $\hat{R}$ and $\hat{S}$ predictions that are illumination and albedo invariant respectively.

over ARAP dataset [11], which consists of realistic synthetic images. We also train the baseline model for additional epochs to get CGIID++ for a fair comparison. Tab. 7 shows different measures to compare two variations of our method with CGIID and CGIID++ on the ARAP dataset [11]. We see improvements in MSE and SSIM scores which compare dense-pixel wise correspondences CSM scores [23] which evaluate R-S disentanglement. Specifically, $CSM_S$ measures albedo invariance of S and $CSM_R$ measures illumination invariance of R predictions. We report CSM scores in two concept set settings of Synthetic vs Real-World from Gupta et al. [23]. The improvement in MSE and SSIM scores appear minor quantitatively, but our method performs significantly better in terms of CSM scores. It also observes superior performance qualitatively, as seen in (Fig. 7)*. Our $R$ outputs are less sensitive to illumination, with most of the illumination effects captured in $S$.

**Discussion and Limitations:** Our concept distillation framework can work on different classification and reconstruction problems, as we demonstrated. Our method can work well in both zero-shot (with concept sets) and few-shot (with bias-conflicting samples) scenarios. Bias-conflicting samples may not be easy to obtain for many real-world applications. Our required user-provided concept samples can incur annotation costs, though concept samples are usually easier to obtain than bias-conflicting samples. When neither bias-conflicting samples nor user-provided concept sets are available, concept discovery methods like ACE [21] could be used. ACE discovers concepts used by the model by image super-pixel clustering. Automatic bias detection methods like Bahadori and Heckerman [6] can be used to discover or synthesize bias-conflicting samples for our method. Our method can also be used to induce prior knowledge into complex reconstruction/generation problems, as we demonstrated with IID. The dependence on the teacher for conceptual knowledge could be another drawback of our method, as with all distillation frameworks [25].

**Societal Impact:** Our method can be used to reduce or *increase* the bias by the appropriate use of the loss $L_C$. Like all debiasing methods, it thus has the potential to be misused to introduce calculated bias into the models.

## 5 Conclusions

We presented a concept distillation framework that can leverage human-centered explanations and the conceptual knowledge of a pre-trained teacher to distill explanations into a student model. Our method can desensitize ML models to selected concepts by perturbing the activations away from the CAV direction without modifying its underlying architecture. We presented results on multiple classification problems. We also showed how prior knowledge can be induced into the real-world IID problem. In future, we would like to extend our work to exploit automatic bias detection and concept-set definition. Our approach also has potential to be applied to domain generalization and multitask learning problems.

**Acknowledgements:** We thank Prof. Vineeth Balasubramanian of IIT Hyderabad for useful insights and discussion. We would also like to thank the four anonymous reviewers of Neurips, 2023 for detailed discussions and comments which helped us improve the paper.

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
