# OpenReview forum: "Concept Distillation: Leveraging Human-Centered Explanations for Model Improvement"
_NeurIPS.cc/2023/Conference — NeurIPS 2023 poster_

### Official Review · Reviewer_YYKb · 2023-07-01

**Soundness:** 3 good
**Presentation:** 3 good
**Contribution:** 3 good
**Rating:** 7
**Confidence:** 3

**Summary:**

The authors introduce an idea by which a student model can become (de) sensitive to some human-understandable concept in its decision-making process. They find CAVs in a teacher model and then transform those CAVs into the student model feature space and use orthogonal vectors to these CAVs to penalize/incentivize models to make decisions based on those concepts.

They improve over baselines on biassed datasets where a concept is known to be the underlying bias.


**Strengths:**

+ I like their idea of debiasing the student model based on human feedback on biased and unbiased samples.
+ method shows superior improvement over existing results when evaluated on the biassed dataset.
+ They visually validate that their model is focusing on more important parts of the images in decision-making.

**Weaknesses:**

+ I'd expect to see some more analysis of the student's model performance. For example, why does the model still get relatively bad accuracy on the color MNIST? is it still biased toward the colors? can we still debias this model and push to get around 100 accuracies?
if not, what are the limitations?

+ This is a minor weak point but I expect to see some recent but relevant related work in the paper. As the paper is considering an alignment between concepts in different models' features spaces, it reminded me of a recent paper "Text-To-Concept (and Back) via Cross-Model Alignment" where an alignment between features spaces is used to interpret a model's behavior in terms of human-concepts. The general framework is somehow similar to this paper.
Furthermore, the idea of considering the normal vector orthogonal to the decision-making half plane is somehow relevant to this paper "Distilling Model Failures as Directions in Latent Space". I want the authors to consider those and other relevant papers.

+ quality of figures should improve (Fig 4).

**Questions:**

My questions are found in the weakness part.

---

> ### Author Rebuttal · Authors · 2023-08-09
>
> We thank the reviewer for very positive feedback. We will address the points below:
>
> >  Analysis on student model's performance
>
> The model remains biased due to the severe bias (100%) in the training set, though we improve significantly over prior efforts. This demonstrates the effectiveness of using explainability measures like CAVs to improve a model. One way to completely debase a model on a chosen concept could be to re-destil iteratively, using results from previous iteration as input to the next. Bootstrapping models in such a way will be interesting and fruitful directions for the future.
>
> > Regarding Related work
>
> Thank you for pointing to the recent papers. We will add them to final version if given a chance.
>
> "Distilling Model Failures as Directions in Latent Space" uses SVMs to find directions of bias in a shared image and language space (of CLIP), then trains using bias-conflicting samples generated by models like Dall-E. Training on bias conflicting samples might not always be feasible due to higher annotation and computation costs. One major difference is that their work proposes data augmentation as debiasing strategy whereas we directly manipulate the gradient vectors which is more interpretable.
>
> "Text-To-Concept (and Back) via Cross-Model Alignment" is a contemporary work (released after Neurips submission) and is very relevant. It maps the activation spaces of two different models using the CLIP latent space similarity. Due to the generality of the CLIP latent space, this approach is useful to encode certain concepts like ‘cat, dog, man’ etc, but it is not clear how it will work on abstract concepts with ambiguous definitions like ‘shading’ and ‘reflectance’ as seen in the IID problem. We are glad to see such current research and think our work will interest many more.
>
> > quality of figures should improve (Fig 4).
>
> We will improve the resolution and clarify the details in all the figures. Thank you!

---

### Official Review · Reviewer_FYuP · 2023-07-06

**Soundness:** 3 good
**Presentation:** 3 good
**Contribution:** 3 good
**Rating:** 6
**Confidence:** 3

**Summary:**

This paper proposes a methodology for training a model to sensitize or desensitize a specific concept. Particularly, it introduces a concept distillation loss that utilizes Concept Activation Vectors (CAVs) derived from a high-performing teacher classifier with abundant knowledge of the concept, aiming to reduce concept activation in that direction. Additionally, it introduces a prediction loss term based on class prototypes to incorporate global features. The paper also employs an autoencoder to minimize the discrepancy between the latent spaces of the teacher and the student. Through experiments on various biased datasets, it demonstrates the robustness of the concept-distilled student model against biases.

**Strengths:**

To the best of my knowledge, this paper is the first to propose the distillation of concept information from a well-trained teacher to a student model. The experimental results provide clear verification that the proposed approach effectively reduces the spurious correlations learned by deep neural networks (DNNs). One particularly interesting result is that even when the teacher model is biased by 100%, the student model still exhibits a reduction in bias. This finding highlights the effectiveness of the proposed methodology in mitigating bias in the student model.

**Weaknesses:**

It seems that one limitation is the use of only simple datasets in the experiments and another is that the overall pipeline is somewhat unclear (please refer to the first question of below section), but other than that, there don't appear to be any specific weakness.

**Questions:**

1) The description of the training pipeline seems somewhat unclear. Has the student model already been trained to achieve a high classification accuracy, and is it now receiving concept knowledge through the prototype loss? If so, wouldn't the latent space of the student model change, requiring constant updates to the module that maps the latent space between the teacher and student models?

2) Regarding the motivation and implementation of concept distillation loss, could it potentially increase the sensitivity to orthogonal concepts as an unexpected side effect?

3) In experiments where there are multiple concepts, how is the loss defined when trying to eliminate bias for all these concepts? Is it defined as an average or debiased one by one?

4) Is it possible to experiment with datasets that have a wider range of concepts? For example, using the Broden concept [1] with ImageNet?

5) Is it feasible to conduct experiments where the activation in a trained teacher model itself is controlled rather than distillation to student?

6) Can additional evaluation be provided to assess how well the autoencoder used for latent space mapping is performing?

[1] Bau, David, et al. "Network dissection: Quantifying interpretability of deep visual representations." Proceedings of the IEEE conference on computer vision and pattern recognition. 2017.

**Limitations:**

It seems that there are no particular specific limitations.

---

> ### Author Rebuttal · Authors · 2023-08-09
>
> We thank the reviewer for his comments and efforts. The observation of the effectiveness of our method in 100% biased teacher is certainly an interesting point. As also explained in our response to Reviewer RoT4y,  we believe this happens as we are defining the concepts using our concept sets (different from the biased training samples) and explicitly making the gradients orthogonal to the CAV during learning.
>
> > Response to Weakness on usage of datasets:
>
> We would like to point that the contemporary literature on debiasing (CDEP, RRR, EG, DFA, EnD, etc) also show results on such datasets (ColorMNIST, DecoyMNIST, BFFHQ). This is because synthetic datasets like ColorMNIST and DecoyMNIST, have extreme and well-defined biases, which are relatively difficult to remove and allow proper analysis of the de-baising method. We additionally introduced a new dataset (TextureMNIST) which is comparatively more challenging and has more real-world bias (textures). Textures being an amalgam of colors and shapes, encompass both scenarios, and CNNs tend to have this bias often as pointed by Geirhos et al.
> Furthermore, Prior-induction in IID using concepts is a novel task attempted in this field. This requires (de)sensitizing to ambiguous and subtle concepts like reflectance-invariance and illumination-invariance. Our method not only effectively (de)sensitizes the model towards the required concepts (Table 4, Figure 7), it does it in a very complex and large network: CGIID (L269, S:L96).
>
> Geirhos, R., Rubisch, P., Michaelis, C., Bethge, M., Wichmann, F. A., & Brendel, W. (2018). ImageNet-trained CNNs are biased towards texture; increasing shape bias improves accuracy and robustness. arXiv preprint arXiv:1811.12231.
>
> We will now answer all the questions one by one below:
> *  RQ 1: *Pipeline:* We start with a pretrained student network which is trained on the biased training set like ColorMNIST (namely Vanilla network) and thus performs extremely poorly on the unbiased test set with out-of-distribution samples like MNIST.
> Eg, Vanilla network gets ~0% accuracy in ColorMNIST, 52.84% in DecoyMNIST, and 11.23% in TextureMNIST, as shown in Table 1.
> We map the pretrained teacher’s activation space to this biased student’s space ONLY for the concept set samples. Mapping step is needed only for CAV projection into the student’s space.
>
> In summary, our pipeline is:
> 0: Mapping teacher space to student space by training the Auto-encoder.
> 1: CAV Learning using distilled teacher outputs to define concepts, etc.
> 2: Training for the main task with Concept Distillation. In this step the top part of the system is not used at all
> 3: Testing or application where the trained model is applied. Only the bottom half of the bottom box is used.
> We will clarify the same in our paper description and algorithm.
>
> *Latent Space Changes:* Yes, the latent space indeed changes gradually with the student’s training but as our distillation converges very fast (in less than an epoch i.e 200-500 iterations), this empirically never observed the drift in any of our experiments.  We tried updating CAV every few iterations (50, 100, 200, etc.), and it made no difference at all. Hence we chose constant CAV in all our experiments.
>
> * RQ 2: During the initial design phase of our method, we chose the concept and its opposite for CAV estimation (e.g. young women vs old women). We updated this to: concept vs random set, in order to specifically avoid the concern being raised here (e.g. young women vs women and men of all ages; see L244-249). This design choice addressed the possible issue of unexpected side-effects, and we empirically confirmed the same.
> * RQ 3: It can be defined either way. While we didn't experiment the effect of mutiple biases in classification problems explicitly, we did so in the case of IID. We tried two scenarios in our IID experiments:
>     1) Setting_1 (L265) wherein branch S was frozen while branch R was being trained for concept of illumination invariance and vice-versa. This represents the case of sequential bias removal.
>    2) In Setting_2 (L266) wherein both R and S branches were trained together along with their common backbone and losses were simply weighted and added.
> For the IID problem, we did not observe significant performance differences. Exploring the same for more than two concepts and in multiple problems is one possible extension that we plan to explore in subsequent extension of our work.
> * RQ 4: We have tried to demonstrate the same with the IID problem (which represents a different task than classification and is trained using a large training set of CGIID with approx. 100k+images). Instead of opting for multiple concepts from [1] for a relatively well-defined problem in classification, we opted to focus on more ‘complex concepts’ like illumination-invariance in an ill-defined problem like IID to gauge the utility of the technique in a more real-world scenario.  If necessary, we can try to get additional results using the experiment suggested.
> * RQ 5: We performed a similar experiment where the distillation was not used and the CAV's were learned in the same model (without proto-types) and reported the result in no distill’ column in Table 5 (L285). We apologize for the unclear explanation and will improve the writing here.
> For completion, we additionally report 'no distill' accuracies (with proto-types) of our model in our Common Response above.
> * RQ 6: It is unclear how mapping module can be standalone evaluated. We indirectly gauge its performance in the overall framework’s accuracy scores.
> One probable way to evaluate the mapping module could be by observing it’s validation mse (we used a small validation set consisting of concept set samples for early stopping the training of mapping module autoencoder. The mse loss reduced by >50% after few epochs and then stabilized).
> We welcome relevant suggestions and experiments for better evaluation.

---

> > ### Author Response · Authors · 2023-08-20
> > **Further response to RQ1**
> >
> > We added the concept loss vs step values in our response *RQ3 to Reviewer gigg* which further substantiates our claim in *RQ1: Latent Space Changes* by Reviewer FYuP.

---

> > > ### Comment · Reviewer_FYuP · 2023-08-20
> > >
> > > Thank you for the response. I think the authors adequately addressed the raised concerns with the explanation and additional experiments. I hope to see the revised paper reflect the feedbacks and responses. I raised my rating.

---

### Official Review · Reviewer_gigg · 2023-07-06

**Soundness:** 2 fair
**Presentation:** 3 good
**Contribution:** 2 fair
**Rating:** 6
**Confidence:** 3

**Summary:**

The paper presents the idea of concept distillation from a pretrained teacher to improve a student. They utilize the notion of concept activation vectors (CAV) as concept representations and adapt a pretrained student to sensitize or desensitize a student w.r.t a concept. They apply this idea for two applications to improve a model in ante-hoc fashion: (i) reducing biases in a student towards a concept, and (ii) incorporating prior knowledge

**Strengths:**

1. The paper is reasonably well written

2. The selection of baselines is quite extensive and the proposed method generally performs well

3. The core idea of using CAV as representations to improve a model indeed has wide applicability. This is shown quite well by authors to a good extent

**Weaknesses:**

I unfortunately have multiple methodological and experimental issues with the. Please find them detailed in the "Questions" sections. The paper has lot of the hallmarks to be a useful and effective work, but I simply didn't enjoy reading it much as I should have, being constantly riddled with doubts in my mind.

**Questions:**

Doubt #1

1. (Methodological, major) I am still not convinced by the use of teacher for CAV estimation. The first supporting argument was in line 170-172 "Direct CAV learning ... as the model may not have sufficiently rich comprehension of the relevant concepts". But even when using teacher representations you map them to the student activation space. So if the mapping $M$ to student activation and back worked perfectly, would it defeat the point of the original argument? The only other supporting argument I could find was an experment on ColorMNIST (Tab. 5). It was nice to see but could it be just due to a specific student architecture. Does this observation generalize to other tasks and more complex students?

2. Can the method be used to improve biases in the original teacher given that you know it has good representation for the concepts? I was assuming that to be able to perform self improvement without a teacher would be more useful.

3. Does the students original activation space get disturbed due to the modifications in its updates or are its layers before the selected intermediate layer fixed? If yes, could it harm the concept distillation loss since the student representations change?

4. (Experimental) Can the model be used to remove many biases simultaneously? This would really enhance the applicability and power of the method.

5. (Minor) Are $M$ and $M^{-1}$ mappings really exact inverses? In that case, isn't $L_{M^{-1}}$ guaranteed to be 0? If yes, then what is the point of $L_{M^{-1}}$? If not, then the notation should be modified.

6. (Minor) I am slightly dissappointed with only one non-MNIST data for debiasing and one for IID task. This is not the biggest concern for me since you do test your method in multiple different ways. And it certainly won't be the deciding factor in my score but nevertheless the impact would feel greater if one of the task was more extensive in its coverage of complex datasets. And I assume you want debiasing as the main task.

**Limitations:**

The authors do discuss in the main paper their limitations. They are also upfront about potential societal impact. I am pretty much satisfied with the discussion.

---

> ### Author Rebuttal · Authors · 2023-08-09
>
> We are glad the reviewer appreciated our core idea and its wide applicability and about it having "the hallmarks to be a useful and effective work." Our main focus is to explore the feasibility of using explainability ideas like concepts for model improvement in multiple use cases. We will improve the writing to convey this message more clearly in the revised draft and answer the specific questions below:
> * RQ1: In theory, it is true that if the mapping module ($M$) had a zero-loss, it could make the distillation case same as the case without distillation but this is not observed in our experiments due to two main reasons: 1) We use $M$ to map ONLY the conceptual knowledge as CAV and train it only for concept sets and not the training samples (L179, Algo 1 L2, Fig 3). 2) Due to major differences in the perceived notion of concepts in teacher and student networks and due to a simple $M$ (one upconv and downconv layer; S:52) the mse loss never goes to zero (e.g in colorMNIST it starts from ~11 and converges at ~5 for DiNO teacher to biased student alignment).  In our initial experiments, we had tried bigger architectures (resnet18+) for $M$ and found improved mapping losses but decreased student performances. We thank the reviewer for this question and will mention it in the paper.
> $M$ encodes an expert’s knowledge into the system via the provided concept sets,  quantifies this knowledge as CAV via a generalized teacher model trained on large amount of data, and thus helps in inducing it via distillation into the student model. This brings threefold advantages in our system: expert’s intuition, large model’s generality, and efficiency of distillation.
> The advantage of distillation in colorMNIST is specifically highlighted in Table 5 with 'no-distill' ablation (also shown in additional ablations in RQ2.). Similar observation can be made for BFFHQ and IID (which have different student architectures, datasets biases, and tasks). Specifically, in MNIST variants we used a two layer convolution network (S:L63), while in BFFHQ, we experimented with ResNet18 as the student backbone (S:L65) while in IID we have a custom two-branch network (CGIID's network, L:269, S:L96). In all these cases, we observed distillation to be substantially more effective than the no-distill case.
> * RQ2: Yes, our method can be used to improve bias in the teacher network. We did the experiment where the distillation was not used, and the CAV's were learned in the same model and reported the result in no distill columns in Table 5 (also L285). This experiment was done with basic concept loss Lc and without the use of proto-types i.e. 'no-distll minus prototypes’. We report an additional ablation 'no-distill plus prototypes' accuracies of our model below.
> | Dataset       | Vanilla        | Without teacher | With teacher |
> |---------------|----------------|----------------|---------------------|
> | ColorMNIST    | 0.1            | 26.97          | 41.83               |
> | TextureMNIST  | 11.23          | 38.72          | 48.82               |
> | BFFHQ         |       56.87      | 59.4           | 63.00                 |
>
> The experiment shows the effectiveness of having the teacher model and highlights the advantage of the rest of the framework, which occurs due to the threefold advantages described above.
> * RQ3: Yes, the student's original space might get changed but it does not impact the overall performance.
> To verify this claim, we did an experiment by updating CAVs in the student after every few training iterations (50, 100, 200, etc.). We observed no significant performance changes due to this. We found that this is because the student's activation space changes gradually relative to the distillation which converges very fast (in less than an epoch i.e 200-500 iterations).
>
> * RQ4: Yes, this is possible. Theoretically, we see no reasons our method cannot be used to remove many biases simultaneously. Training for IID (both R-S training experiment, L265) we have two concepts: Reflectance's illumination-invariance and Shading's color-invariance, which are optimized simultaneously in the model. Exploring the same for more than two concepts and in multiple problems is one possible extension that we plan to explore in subsequent extensions of our work.
> * RQ5: No, they may not be the exact inverses. They were meant to denote the reversed mapping spaces of the encoder and decoder only (L177). Sorry for the confusion and we will change the notation to Encoder E and Decoder D.
> * RQ6: Instead of opting for multiple concepts for a relatively well-defined problem in classification, we focussed on more complex concepts like illumination-invariance in an ill-defined problem like IID to gauge the utility of the technique in a more real-world scenario.
> We would like to point our that MNIST datasets have been chosen because of two reasons:
>     1) Synthetic biases in MNIST datasets are designed to be extreme and hence more challenging than a real world scenario. Furthermore, the introduction of abstract biases in a well-understood dataset helps gauge the impact of the technique in a better way.
>     2) MNIST allows comparisons with other contemporary works, which all report using the same methodology.
>
>    Apart from this, we would ike to point that we have introduced a more challenging TextureMNIST dataset and also report results on BFFHQ, which has natural images. In case of IID, the IIW benchmark is the only large scale human annotated dataset (with rest of the datasets like Sintel, being synthetically rendered with known issues in the IID evaluation). As pointed out by the reviewer, we focused on gathering evidence for our method’s utility for multiple tasks and various types of biases rather than multiple datasets for a similar task. If needed, we can include additional results in the revision.
>
> We once again thank the reviewer for the appreciation of the work and its utility and hope his/her reservations are addressed sufficiently.

---

> > ### Comment · Reviewer_gigg · 2023-08-13
> >
> > I want to thank the authors for the rebuttal.
> >
> > The results for ''no-distill" case for all datasets apart from ColorMNIST should certainly added in the main paper/appendix to support your design choices.
> >
> > I still have a few remaining questions:
> >
> > (1) RQ1 - How would you describe a practitioner can empirically assess if the mapping module is functioning as desired? Your response to reviewer FYuP in RQ6 initially seemed fair enough practical choice to me, i.e, "validation mse reduces by a certain amount and the loss stabilizes". But if validation mse drops highly would you then think that $M$ might be causing decrease in student's performance? If yes, then is the optimal functioning of $M$ indicated by the student's performance rather than the validation mse?
> >
> > (2) RQ2 - If I understand correctly, "no-distill" results don't use a teacher and directly debias student architecture. Did you also try to debias the teacher architectures? The reason I am making this distinction is because the teacher architectures are assumed to have richer capacity to encode conceptual knowledge.
> >
> > (3) RQ3 - From what I understood from Alg1, $M$ and $v_C^l$ are learnt before student updates and fixed throughout student training. Please correct me if I am mistaken in this understanding (it is important!). If this is indeed true I fail to see how your experiment verified your claim in RQ3? The experiment you describe seems a more sensible way of training, compared to keeping $v_C^l$ fixed throughout student updates. What I am interested to see instead is the value of $L_C(x)$ with the new student CAV from the end of training. Is the loss gradient still orthogonal to the new CAV while it was trained the whole time with old CAV?

---

> > > ### Author Response · Authors · 2023-08-15
> > >
> > > We thank the reviewer for his comments and will address them one by one below:
> > >
> > > * **RQ1**: We concur with the observation that validation MSE is not the best way of evaluating the mapping module’s performance, as we wrote to FYuP: “We indirectly gauge its performance in the overall framework’s accuracy scores.” Additionally, the reduced student sensitivity can be gauged by our reduced concept loss (given in RQ3 below) as well as TCAV scores [Kim et al.] reported below for vanilla network: "Student (before training)" and "Student (after training)" (essentially "Ours" in Table 3 and Table 5)
> > >
> > > ### TCAV Scores
> > > |     Dataset      | Student (before training) | Student (after training) |
> > > |-----------|--------------------------|-------------------------|
> > > | ColorMNIST|           0.52           |          0.21          |
> > > |   BFFHQ   |           0.78           |          0.13          |
> > >
> > > The evident reduction in TCAV scores (particularly for the concepts of *color* in ColorMNIST and the concept of *age* in BFFHQ) subsequent to training signifies our method's efficacy in mitigating bias within the models which is also demonstrated by improved accuracies in the paper (Table 3, 5).
> > > * **RQ2**: We operate under the scenario of leveraging a pre-trained teacher and assume no access to its training regime. We assume access only to the student's training dataset/losses/etc.  Debiasing the teacher is not in scope as a result. Better teacher models will become available in the future, and we should be able to take advantage of them right away.
> > > We would like to clarify the term “richer” used with respect to the teacher. Teacher models like DINO are large vision models that are trained on hundreds of millions of images and have the capacity to be used in different applications and settings. That’s what we meant effectively in L:170-173 and in Supp L:30.  We will clarify this point in the paper to avoid any confusion.

---

> > > > ### Author Response · Authors · 2023-08-15
> > > >
> > > > * **RQ3**: Yes, we keep CAV $v_{C}^l$ and $M$ fixed in student updates, this is because experimentally there were no improvements when we varied the CAV's for the reasons mentioned before. We will modify the algorithm description to add the $M$ re-learning and CAV updation steps for the sense of completion.
> > > >
> > > > We show the training $L_c(x)$ values in two settings: when CAV is updated every 200 iterations *cav_update_iter200* and when CAV is kept constant *constant_cav* on ColorMNIST (for the case of "no distill" cav learning in student model) below:
> > > >
> > > > | iters | $L_c(x)$ in cav_update_iter200 | $L_c(x)$ constant_cav |
> > > > |-------|------------------------------|-----------------------|
> > > > |   0   |            105.98            |        106.048       |
> > > > |   10  |            94.09             |        94.396        |
> > > > |   50  |            60.96             |        60.328        |
> > > > |  100  |            36.28             |        25.27         |
> > > > |  150  |            14.11             |        10.37         |
> > > > |  190  |            10.18             |        8.153         |
> > > > |  198  |             9.99             |        8.526         |
> > > > |  **199**  |             **9.51**             |        **7.811**         |
> > > > |  **200**  |    **20.61 (cav updated)**       |        **7.508**        |
> > > > |  210  |            17.010            |        7.484         |
> > > > |  250  |            10.94             |        5.119         |
> > > > |  300  |            25.27             |        5.285         |
> > > > |  350  |            12.46             |        4.96          |
> > > > |  390  |            11.81             |        4.417         |
> > > > |  **399**  |            **11.72**            |        **4.359**         |
> > > > |  **400** |    **31.58 (cav updated)**       |        **4.434**         |
> > > > |  ...  |            ...               |        ...          |
> > > > |  450  |            11.30             |        3.908         |
> > > > |  490  |            9.890             |        3.879         |
> > > > |  499  |             8.90             |        3.577         |
> > > >
> > > > As seen above, there is a recurring pattern with $L_c(x)$ increasing on CAV update (due to abrupt change in objectives), followed by a subsequent decrease due to optimization. The loss, however, remains lower than the initial value the model started with (from 105.98 at iteration 0 loss dropped to 9.51 at iteration 199 (before CAV update) but jumps to 20.61 following CAV update), underscoring the efficacy of our approach. Similar patterns were consistently observed when updating CAVs in varying numbers of iterations. This trend persisted across diverse datasets during training as well.
> > > >
> > > > Furthermore, aligning with the organizers' advice regarding image attachments, we've included a graph depicting the relationship between $L_c(x)$ and iterations in our comment to AC. The graph encompasses multiple settings:
> > > > 1) *cav_update_iter200*: CAV updates every 200 iterations.
> > > > 2) *cav_update_iter100*: CAV updates every 100 iterations.
> > > > 3) *constant_cav*: Fixed CAV throughout training.
> > > >
> > > > The given graph is shown for experimentation over ColorMNIST, but we find the same recurring $L_c(x)$ patterns across all other datasets. Additionally, the best validation accuracy (and also corresponding test accuracy) values for all the settings mentioned above (whether fixed or varying CAVs) are the same (within < 0.3% accuracy changes amongst the settings). The graph also shows that our design choice of keeping CAVs fixed is good practically as the loss *quickly* converges to a low value.
> > > >
> > > > We thank the reviewer for these comments/suggestions. Our use of an interpretability method of concepts to (de)sensitize the base model does have many directions that need exploration, including these and others.

---

> > > > > ### Comment · Reviewer_gigg · 2023-08-15
> > > > >
> > > > > Thank you for the reply. I appreciate your responsiveness. I would recommend adding the TCAV scores for all datasets in the main paper, even if as a brief mention. It is a another quantifiable indicator to demonstrate debiasing, apart from the accuracy. Secondly, the CAV update should also be incorporated in the algorithm. I would consider it as an important detail.
> > > > >
> > > > > Since I don't have anymore major concerns and I believe the work has sufficient positives, I am updating my score from 4 to 6.

---

> > > > > > ### Author Response · Authors · 2023-08-16
> > > > > >
> > > > > > Thank you for pushing us to dig deeper into all aspects of the method. The work and the presentation have improved as a result.

---

> > > > ### Author Response · Authors · 2023-08-20
> > > > **Further response to RQ1**
> > > >
> > > > We did an experiment to evaluate the mapping module empirically in response to *RL1* by Reviewer oT4y and the response is reproduced here since Reveiwer gigg asked this in *RQ1*.
> > > >
> > > > **Further Response to RQ1:**
> > > > Apart from checking TCAV scores of student as reported earlier, another way of checking the effectiveness of mapped CAV empirically is via cosine similarity of the concept images to the CAV (like Kim et al.). Kim et al. check the quality of CAV by sorting concept images according to their cosine similarity wrt to the learned CAV. We measured the cosine similarity of concept images with corresponding CAV for the three networks: teacher, mapped teacher, and student. For example, in ColorMNIST zeros are always associated with the color "red". We learn a separate CAV for concept red (CAV_red) in each of teacher, mapped teacher and student and measure its cosine similarity (cs) with the respective model representations for concept images of red, red-zeros and found these trends:
> > > >
> > > > #### Cosine Similarity Order of concepts with CAV_red
> > > > *Teacher:*  **cs(red) >  cs(red-zeros)** < cs(red-non-zeros)  | Indicates correct concept learning
> > > >
> > > > *Mapped Teacher:*  **cs(red) > cs(red-zeros)** <  cs(red-non-zeros)  | Indicates correct concept learning
> > > >
> > > > *Student:* **cs(red) <  cs(red-zeros)** < cs(red-non-zeros) | Indicates confusion of bias with concept (concept red-zeros confused with concept red)
> > > >
> > > > As seen from the above ordering, the teacher's CAV and its mapped version capture the intended concept well while the student confuses the red-zeros concept with CAV.
> > > > This small demonstration shows (a) why the teacher is needed for good CAV learning and (2) how well CAV's are transferred to the student via the mapping module. Mapping module does not bias the CAV representation.
> > > >
> > > > We will further add the qualitative cosine similarity-based sorting results to the paper/supplementary.

---

### Official Review · Reviewer_oT4y · 2023-07-06

**Soundness:** 2 fair
**Presentation:** 2 fair
**Contribution:** 2 fair
**Rating:** 3
**Confidence:** 4

**Summary:**

This paper aims to sensitize or desensitize a (smaller) student model with respect to user-provided high-level concepts, by leveraging a (larger) teacher model. Specifically, a supervised mapping model learns a bijection between some chosen latent space in the teacher model and the student model. Then, CAVs extracted from teacher model is mapped to the student model through the mapping model. Finally, the student model is trained to sensitize / desensitize with respect to the mapped CAVs by minimizing / maximizing the concept sensitivity score.

**Strengths:**

* The ideas in this paper are quite creative and have plenty of potential if explored properly.
* Working with new real-world datasets (i.e. IID experiment) in the interpretability domain is always welcomed, which makes interpretability methods more application grounded.

**Weaknesses:**

Overall my impression is the paper explores too many ideas at once and thus unfortunately fails to cover each one in detail. Design choices are not well-justified and phenomena are not well-understood.

* I find this paper extremely hard to read. Ideas are not elaborated clearly when first introduced and pieces of information are scattered all over the paper. It takes about 3x the time to read this draft compared to other ones. Perhaps the writing needs to be cleaned up.
    * Example 1: In P8 the ablation for ColorMNIST is placed after IID experiments which is confusing.
* The proposed method is too complex and has too many claims that are not well substantiated.
    * Claims:
        * Optimizing for concept distillation loss (Eq 2) can effectively sensitize / desensitize
        * Prototype-based loss captures sensitivity better at a global level and "facilitates the use of any intermediate layer by serving as pseudo-class label"(P5L168) which frankly I have no idea what this means.
        * Large models learns CAV better than smaller ones. (the entire premise for adopting a teacher-student framework)
* The compared baselines may not necessary be fair since the proposed method is a fully-supervised one that requires training of entire models while baselines are either zero-shot or few-shot. Should compare with fairness/debiasing methods that requires full training of the entire model.

**Questions:**

* For each of the claims, there should be dedicated experiments to verify the claims, instead of bunching everything together and comparing this menagerie method with weaker 0-shot/few-shot baselines. Please justify the claims explicitly.
* A proper ablation study on the design choices is a necessary component for such complex method. How does each component affect the result?
* For the concept distillation loss, the entire student model is optimized for Eq (2) but the (mapped) CAV is kept constant. If layers prior to the latent space where the CAV lies is modified, CAV no longer represents the latent space because the input representation has changed. For example, suppose some concept happens to lie in the first dimension of the CAV latent space. If we optimize the entire model, the model could switch the concept and encode it in the second dimension instead. Thus, does it not make sense to only optimize for the model after the CAV layer?
* There are no descriptions of what each entry in Table 6 represents. P9L288 states to "See supplementary for detailed explanations" but I haven't been able to find where them. Can the authors point me to the exact section for the ablation study?
* Why exactly is $M^{-1}$ needed?
* Can the authors elaborate on why concept distillation from a completely biased teacher AND the teacher shares the same architecture as the student model might still improve? The result is quite surprising considering the benefits of distillation has disappeared in this setting.

**Limitations:**

There is a small section in P9L299-302. The only limitation stated is "The dependence on the teacher for conceptual knowledge could be another drawback of our method as with all distillation frameworks". One limitation might be when CAV extracted from the teacher model fails to translate to the student model. When, where, and why does this happen?

---

> ### Author Rebuttal · Authors · 2023-08-09
>
> We thank the reviewer for the appreciation of our idea’s potential and novelty including demonstration on real-world dataset on a reconstruction problem.
>
> The primary goal of this work is to show that explainability ideas like concepts can be used in a loop to improve the models. We believe it is an important idea with many extensions as observed by all reviewers. We apologise if the writing is confusing and will improve it if given a chance.
>
> We now address the concerns:
> * W1: We placed all experiments (the evidence) first followed with ablations at the end as done by most. We can include references earlier.
>  * W2: We use concept distillation to desensitise a model to user desired concepts effectively. We introduced a novel concept loss and a prototype-based extension of CAVs. The loss can be used directly in the same model (no distil experiment in Tab 5) or using a teacher which has a richer knowledge of concepts. The prototypes transfer labels for loss computation in the chosen intermediate layers. These components, in our view, combine towards the core idea.
>      * W2.1: Effectiveness of concept distillation on different datasets and different tasks are shown. Improvement in accuracies of models in biased settings is substantial (Tables 1 column "Ours+L" vs Table 3 column "Ours"). The use of concepts leads to better generalization (Table 2, Figure 6). Finally, we demonstrated effectiveness in a challenging real world IID problem where none of the existing interpretability based model improvement methods could be applied ( Table 4, Figure 7)
>      * W2.2 Prototypes have been used widely in the existing literature (L127, [Xue et al,  Keswani et al]) and have proved to capture class-wide characteristics well (L126, 127). They are a type of global method (L31 definition of proto-types, L64 for local vs global) by their very definition (as they are based on feature clusterings). Original CAV based sensitivity representations are limited to the last layer due to their loss based definition (explained in L131-138) and hence cannot be used to estimate the sensitivity to an intermediate layer. To extend CAV sensitivity to any layer $l$, we need a way to calculate model's sensitivity to it, which requires a loss to be calculated in layer $l$ (instead of the final layer where GT annotations are available). We circumvent this issue by estimating prototypes of the GT annotations at any layer (by forward passing and clustering the feature vectors based on labels in the chosen intermediate layer) (L168). This is a natural extension of prototype usage in literature.
>      * W2.3: We experimented with various architectures for teacher selection. Results in supplementary L27, L34 show larger model perform better. Please see no-distill in Table 5 as well as additional ablations in the common rebuttal and also in our response to Reviewer Gigg.
> * W3: *Zero-shot vs few-shot*
> We are sorry we failed to clearly convey the experimental settings to the reviewer. Our method is zero-shot and not supervised. Zero-shot vs few shot are defined wrt training data (L195): zero-shot means the model has never seen the distribution of test samples (Xian et al.), while in few-shot, the model has access to some examples from the distribution of the test set (Wang et al.).  Our method works with abstract concept sets (different than any test sample) and is essentially zero-shot. Our comparison with other zero-shot interpretability methods like CDEP, RRR, EG (Table 1) is justified. Our method can also take advantage of few-shot examples if available as shown in results on BFFHQ (Table 3). In this experiment, our concept set comprises old/young men/women extracted from the test set (thus making it a few shot method as it has seen samples from test-distribution). We compare with few-shot methods EnD and DFA (Table 3) (L114-L117). This also makes our comparison fair in this category as well.
>  *Full training of the entire model:* All of the above methods train the entire model with introduced losses/rules/samples. CDEP, RRR, EG have their loss terms added to the models like us while EnD and DFA train the entire model using their proposed algorithms using the ood samples as well.
>
> * RQ1: We hope we answered all the doubts of the reviewer in our responses above. We will try to rewrite better in the final version.
> * RQ2 and RQ4: We explored the impact of each design choice in several experiments in the ablations section table as well as in supplementary (Table 5, Table 6, Figure 8, L 278 and S:L34-45). We have provided a detailed explanation for the ablations of Table 6 in the common rebuttal above. We apologise this has not been explained more clearly in the paper.
> * RQ3: We will change the notation as mentioned in the common rebuttal too.
> * RQ5: When 100% biased teacher is used, the student gets 23% accuracy on ColorMNIST which is still low compared to 50.93% accuracy when a Dino ViTB/8 is used.
> * RL1: Theoretically, this can happen if the student's space for the chosen bottleneck layer does not have sufficient capacity to represent the concepts (e.g., Teacher's space is huge while students is very small for chosen bottleneck).  But this is unlikely to occur because the student classifier's activation space is expected to have the capacity to encode the detailed images from iid ood samples and perform classification or the task at hand. Concepts in general are supposed to be more abstract compared to these detailed images and should be encoded practically in the student space. This is what we found in our experiments.
>
> We once again thank the reviewer for incisive comments and hope our explanation has made the reviewer positive about our work.

---

> > ### Comment · Reviewer_oT4y · 2023-08-18
> >
> > Many thanks to the authors for the detailed reply. I guess how easily understandable a paper may be quite subjective, although I still believe that the entire paper could be structured better to illustrate the key points. That being said, let's get into the support for each of the claims.
> >
> > W2.1:
> >
> > I now understand the experiment setting better. Can the authors include the results for CAV calculated only using the student model in Table 1 as well (I believe the no-distill results in the author's rebuttal)? It is quite relevant for justifying the usage of a teacher-student framework.
> >
> > W2.2:
> >
> > 1. I don't quite get why the original CAV formulation could only be applied on the final layer. Even the original TCAV paper experimented with TCAV scores calculated in multiple intermediate layers. One simply back-propagates the input gradients to a specific layer of interested and calculate the sensitivity there instead.
> > 2. Prototypes being a popular technique does not directly justifies its usage. I don't believe this type of prototype usage have been explored before in past literature (for CAVs) and would probably require an entire work to examine whether this design choice makes sense. The authors could also point me to the section where they explored the design choice of adding the prototype against other alternatives, as I could not find it.
> >
> > W2.3:
> >
> > Can the authors provide intuition for why a CAV learned by a teacher and then passed to the student via a learned Autoencoder would be better than directly learning the CAV in the student? My understanding is that any additional "rich" information the teacher model contains must be discarded in the process of the Autoencoder mapping. Therefore, any additional discriminative features used for learning the CAV in the teacher model would not be able to translate (through the Autoencoder) into the student model. The information that could be able to be translate (through the Autoencoder) into the student model are already existing in the student model. Thus, there is no information advantage if we learn CAV from the teacher, if the CAV needs to go through the (compressive) Autoencoder.
> >
> > RQ3:
> >
> > The authors didn't respond to whether only finetuning the layers after the selected intermediate layer could circumvent the changing latent space issue (which I believe would and should be the preferred method for debiasing).
> >
> > RL1:
> >
> > One solution is to first check if the concept is translatable from the teacher to the student first is an important step to include in the algorithm. This is an analogy of how TCAV would check the t-test statistics to see if a concept could be represented in a latent space. Confirming whether the distillation is feasible before distilling should be mandatory.

---

> > > ### Author Response · Authors · 2023-08-20
> > > **Response to Reviewer oT4y**
> > >
> > > We thank the reviewer for the comments. We are glad many design options we explored while developing the idea and some new ones are being brought out in this discussion. This will improve the content and writing of the work.
> > >
> > > We will now discuss the comments one by one.
> > >
> > > **W2.1:**   Yes, we will include the mentioned results in Table 1. We additionally show why CAV calculation in teacher is better via an experimentation in RL1 below.
> > >
> > > **W2.2.1:**  The original CAV sensitivity as introduced by Kim et al. can be calculated at any intermediate layer $l$, but they only measured the sensitivity of the final layer prediction wrt activations in $l$. They were interested only in the question: if any changes in activations are done in $l$ what is its *effect on the final layer prediction*?
> > > This is different from the question we ask: if any changes in activations are done in $l$ what is its *effect on any other layer*? We calculate the *sensitivity of $l$ prediction* wrt activations in any layer. To this end, we use proto-types that act as pseudo GT labels.
> > >
> > > One further question could be: Why use intermediate layer sensitivity instead of the last layer?
> > >
> > > Ans: TCAV by Kim et al. was designed to be an interpretability method to check for model sensitivity to certain concepts for classification problems. They estimated the sensitivity using the "final layer's loss/logit".  We aim to finetune the model by (de)sensitizing it towards a given concept that could exist in any layer (S:22-26, [1]) for which we use prototypes based loss ($L_p$) in that layer.
> > >
> > > **W2.2.2:** The answer is linked to our reply to *W2.2.1* above. We believe the use of proto-types with CAV is a novel contribution (L77). As discussed above, this was necessary to compute intermediate layer loss for CAV sensitivity estimation at *any* layer. Defining loss in final layers is straightforward due to the availability of ground truth (GT), which are not available for the intermediate layers. Prototypes were used as pseudo-GT class labels in various classification scenarios [2, 3, 4, 5, 6, 7, 8, etc. (referred earlier)]. We build on that in our experiments. In our opinion, clustering based proto-types [2, 3, 4, 5] presented the simplest way to achieve the same. In our initial experiments, we tried using dimensionality reduction techniques, global pooling, etc., to reduce class activations to a scalar as a substitute for CAV logit (as proposed by Kim et al.). Those yielded unsatisfactory performance.
> > >
> > > The advantage of using intermediate proto-types vs the last layer's logits for the concept (de)sensitization is shown in Table 6 and discussed in the common rebuttal (ablations).
> > >
> > > [1] Akula, A., Wang, S., & Zhu, S. C. (2020). Cocox: Generating conceptual and counterfactual explanations via fault-lines. AAAI
> > >
> > > [2] Caron, M., Bojanowski, P., Joulin, A., & Douze, M. (2018). Deep clustering for unsupervised learning of visual features. ECCV
> > >
> > > [3] Yang L, Huang B, Guo S, Lin Y, Zhao T. A Small-Sample Text Classification Model Based on Pseudo-Label Fusion Clustering Algorithm. Applied Sciences. 2023
> > >
> > > [4] Li, J., Zhou, P., Xiong, C., & Hoi, S. C. (2020). Prototypical contrastive learning of unsupervised representations. arXiv.
> > >
> > > [5] Niu, C., Shan, H., & Wang, G. (2022). Spice: Semantic pseudo-labeling for image clustering. IEEE Transactions on Image Processing.
> > >
> > > [6] Nassar, I., Hayat, M., Abbasnejad, E., Rezatofighi, H., & Haffari, G. (2023). PROTOCON: Pseudo-label Refinement via Online Clustering and Prototypical Consistency. CVPR.
> > >
> > > [7] Tanwisuth, K., Fan, X., Zheng, H., Zhang, S., Zhang, H., Chen, B., & Zhou, M. (2021). A prototype-oriented framework for unsupervised domain adaptation. NeurIPS.
> > >
> > > [8] Li Y, Guo L, Ge Y. Pseudo Labels for Unsupervised Domain Adaptation: A Review. Electronics. 2023.
> > >
> > > **W2.3:** This is essentially the case of ideal mapping with a loss of 0. We train the mapping module *only for concept sets* (and not the training dataset), as discussed in detail in the response RQ1 to Reviewer gigg. We also show an experiment in RL1 below (order of cosine similarity scores) which shows the same. We will mention this case in the paper/supplementary to avoid confusion and enhance comprehension.

---

> > > > ### Author Response · Authors · 2023-08-20
> > > > **Response to Reviewer oT4y ... continued**
> > > >
> > > > **RL1:** Currently, we check the teacher to student CAV transferability indirectly from the improved accuracy (reported in the paper) and TCAV scores. We list the TCAV scores [Kim et al.] below for vanilla network: "Student (before training)" and "Student (after training)" (essentially "Ours" in Table 1 and Table 3)
> > > >
> > > > #### TCAV Scores
> > > > |     Dataset      | Student (before training) | Student (after training) |
> > > > |-----------|--------------------------|-------------------------|
> > > > | ColorMNIST|           0.52           |          0.21          |
> > > > |   BFFHQ   |           0.78           |          0.13          |
> > > >
> > > > The evident reduction in TCAV scores (particularly for the concepts of color in ColorMNIST and the concept of age in BFFHQ) subsequent to training demonstrates our method's efficacy in mitigating bias within the models, along with the increased accuracy reported earlier.
> > > >
> > > > We employed t-testing initially by taking concept vs multiple random samples and selecting only the significant CAVs. This proved to be too expensive computationally during training, especially during frequent CAV updates. Currently, we have a simple filter on CAV classification accuracy > 0.7 to select only the good CAVs (i.e. CAVs that can differentiate concept vs random). The concept loss $L_c$ corresponding to all such valid CAVs is then averaged before backpropagating. This design simplification was empirically verified and found to work equivalently to Kim et al.'s t-testing).
> > > >
> > > > Yet another way of checking the effectiveness of mapped CAV empirically is via cosine similarity of the concept images to the CAV (like Kim et al.). Kim et al. check the quality of CAV by sorting concept images according to their cosine similarity wrt to the learned CAV. We measured the cosine similarity of concept images with corresponding CAV for the three networks: teacher, mapped teacher, and student. For example, in ColorMNIST zeros are always associated with the color "red". We learn a separate CAV for concept red (CAV_red) in each of teacher, mapped teacher and student and measure its cosine similarity (cs) with the respective model representations for concept images of red, red-zeros and found these trends:
> > > >
> > > > #### **Cosine Similarity Order of concepts with CAV_red**
> > > > *Teacher:*  **cs(red) >  cs(red-zeros)** < cs(red-non-zeros)  | Indicates correct concept learning
> > > >
> > > > *Mapped Teacher:*  **cs(red) > cs(red-zeros)** <  cs(red-non-zeros)  | Indicates correct concept learning
> > > >
> > > > *Student:* **cs(red) <  cs(red-zeros)** < cs(red-non-zeros) | Indicates confusion of bias with concept (concept red-zeros confused with concept red)
> > > >
> > > > As seen from the above ordering, the teacher's CAV and its mapped version capture the intended concept well while the student confuses the red-zeros concept with CAV.
> > > > This small demonstration shows (a) why the teacher is needed for good CAV learning and (b) how well CAV's are transferred to the student via the mapping module. Mapping module does not bias the CAV representation.
> > > >
> > > > We will further add the qualitative cosine similarity-based sorting results to the paper/supplementary.

---

> > > > > ### Comment · Reviewer_oT4y · 2023-08-21
> > > > >
> > > > > Many thanks to the authors for the detailed explanation and insightful experiment.
> > > > >
> > > > > W2.2: The additional explanation (significantly) helped me understand the importance of prototypes for capturing concepts in intermediate layers. I don't believe this is trivial for readers and would strongly recommend adding these explanations in the introduction (P1L32). The current intuition for prototypes is lacking. Elaborate that the original CAV is only concerned with the last layer and why extracting concepts in intermediate layers helps extract all the concepts that is embedded throughout the entire network.
> > > > >
> > > > > RL1: The cosine similarity experiment is actually quite insightful and I believe belongs in the main paper. This is the evidence needed to justify why teacher-student might be beneficial for concept extraction. A couple of questions:
> > > > > 1. Can the authors provide the exact numbers for the similarity? The > then < notation is quite confusing.
> > > > > 2. Why does cs(red) > cs(red-zeros) for the Teacher model indicate correct concept learning? Shouldn't all red inputs be similarly correlated to the red concept (CAV_red), despite zeros or not?
> > > > > 3. Including the similarity for "non-red, zeros" would indicate how correlate the zero concept and red concept is mixed together in these three models.
> > > > >
> > > > > One final note regarding ablation study in the original review (RQ2), something that is straightforward on the design choices would be greatly beneficial (as opposed to the ones in Table 5, 6 which also involves other parameters).
> > > > >
> > > > > Method Name | Teacher? | Prototype? | General Performance (check if dropped bc of debiasing) | Protected attribute performance (check if increased bc of debiasing)
> > > > > --- | --- | --- | --- | ---
> > > > > CAV | X | X | ??? | ???
> > > > > Ours (but only on last layer like CAV) | O | X | ??? | ???
> > > > > Ours (prototype on student only) | X | O | ??? | ???
> > > > > Ours | O | O | ??? | ???
> > > > >
> > > > > The information for this table may already exist in the main paper but presenting it in a more direct manner would be greatly beneficial for understanding this work. Table 6 is extremely hard to read and understand. I would strongly suggest placing this table in the early portions of the paper to help readers build intuition (e.g. Table 1).

---

> > > > > > ### Author Response · Authors · 2023-08-22
> > > > > >
> > > > > > We thank the reviewer for pushing us further. We really appreciate it.
> > > > > >
> > > > > > **W2.2:** Yes, we will be elaborating the mentioned details in paper more clearly as suggested.
> > > > > >
> > > > > > **RL1:** Yes, we will add ths experiment to main paper.
> > > > > >
> > > > > > **1.** Here are the exact additional numbers:
> > > > > >
> > > > > > | Model           | 'red'   | 'red_zeros' | 'red_non_zeros' | 'non_red_zeros' | 'inverted_color_zeros' |
> > > > > > |-----------------|---------|-------------|-----------------|-----------------|-------------------|
> > > > > > | Teacher         | 0.084   | -0.013      | -0.009         | 0.005           | 0.006             |
> > > > > > | Mapped Teacher  | 0.287   | -0.044      | 0.014          | 0.024           | -0.037            |
> > > > > > | Student         | -0.023  | -0.019      | -0.013         | 0.000           | -0.038            |
> > > > > >
> > > > > > Please note that numbers across the rows are not comparable, but across the columns are. Teacher is in a different activation space (with dimensions: 64x3x3 (channel first)) while Mapped Teacher and Student are in the same activation space (with dimensions: 50x8x8). Hence teacher is not directly comparable to Mapped Teacher and Student. Teacher has significantly high score of concept 'red' compared to its score of other concepts (in row1). Also 'red_zeros' and 'red_non_zeros' have lower score than 'red' (for reason explained below in 2.) but higher scores than that of non-red concepts ('non_red_zeros', 'inverted_color_zeros')
> > > > > >
> > > > > > **2.** Concept red is evaluated using patches of red colour against other colours. Concepts like 'red_zeros' and 'red_non_zeros' have additional concepts: use a mix of colours and digits, with 'red' digit in black colour background in common. Thus, cs(red) > cs(red_zeros) indicates learning red better. The main point is that the teacher and mapped teacher have clearly learned the red concept over others as seen by the relative values.  The vanilla network of the student confuses red with red_zeroes and red_non-zeros.
> > > > > >
> > > > > > **3.** As suggested, we added the scores of 'non_red_zeros'  above.
> > > > > > We also added another concept set 'inverted_color_zeros' which has the color inverted from ColorMNIST training set (i.e., ColorMNIST test set).  All three models have really low scores of  'non_red_zeros' and  'inverted_color_zeros' with CAV_red.
> > > > > >
> > > > > > Interestingly student has a high score of CAV_red with 'red' while a lower score for 'non_red_zeros'. This can be explained by the way the student is trained: it always sees red to predict zeros, but not otherwise. Hence red is associated with zeros (high cs( 'red_zeros' )) but not zeros with red (low cs('non_red_zeros').
> > > > > >
> > > > > > We thank the reviewer for the effort taken to suggest how the ablation can be presented. We will certainly do that.

---

### Author Rebuttal · Authors · 2023-08-10

We thank reviewers for valuable suggestions and feedback. We are glad they acknowledged the potential of our work. Our primary goal is to show explainability ideas like concepts can be used in a loop to improve the models. This is an important idea with many future directions as observed by reviewers. The summary of our pipeline (Fig 3 and Alg 1):
* Step 0: Mapping teacher space to student space by training the autoencoder.
* Step 1: CAV Learning using distilled teacher outputs to define concepts.
* Step 2: Training the main task with Concept Distillation. The top part of Fig 3 is not used
* Step 3: Testing where the trained model is applied. Only the bottom half and classification head are used.

We address common concerns here. Detailed responses are included against reviews.

**Mapping module Notation & Training:**
* M and M-1 notation are meant to show spaces of encoder and decoder only (L177). They are not exact inverses.  We will change it to Encoder E and Decoder D.
* We use the mapping module ONLY for the concepts while training for CAV. It has no role in later steps like concept distillation (L179, Algo 1 L2, Fig 3).

**Ablations:**
Several ablations to justify our design choices (Tabs 5, 6; Fig 8). We tried our concept loss formulation in no distillation setting wherein the CAVs are learned in the same student model (table 5, no distill). The improvement in accuracy goes from 0% in base vanilla network trained with Lo to 9.96% when Lc (eq 2, L147) is added during the training. Sensitivity used for Lc  is calculated as gradient of loss (eq 2 L147) in our experiments. We tried by calculating sensitivity as the gradient of logit (second column of table 6, "logit") and found the former works better.  We also ablate using a variant where we compute our concept distillation loss using last layers loss directly (the way Kim et al[28] calculated sensitivity) instead of using our proposed proto-type based loss, i.e. calculation of Lc, using Lo instead of Lp described in L155 (Lo column). This shows the effectiveness of using mean class representations (proto-types).

We ablate using a fixed set of prototypes instead of varying them (Lp, fixed proto) and found the performance to reach 40.02%, by varying the proto-types according to L164 we achieve 41.83% accuracy which is a slight improvement over fixed proto-types case. Finally, we add a local loss (RRR loss term) and achieve 50.93% accuracy on colorMNIST. We also show results when i) KNN K in proto-type calculation is varied and found k=7 to work best (Figure 8 Right) ii) number of images (# imgs) in the concept set are varied and we observe a peak in # imgs = 150 which is chosen as the KNN K and # imgs in our experiments (L189, S:61, S:74)
We apologize for the unclear ablations and will add them to the paper to make them clear.

**No Distillation:**
We showed an ablation in main paper (Tab 5 L285)) where the distillation was not used and the CAV's were learned in the same model (without proto-types) and reported the result in 'no distill’ column in Tab 5 (L285) which is 9.96%. We additionally report 'no distill’ accuracies (with proto-types) of our model below:
| Dataset       | Vanilla        | Without teacher | With teacher |
|---------------|----------------|----------------|---------------------|
| ColorMNIST    | 0.1            | 26.97          | 41.83               |
| TextureMNIST  | 11.23          | 38.72          | 48.82               |
| BFFHQ         |       56.87      | 59.4           | 63.00                 |

These results are much better than Vanilla network even without teacher but much better with it.

**Latent Space:**
The student's original space might get changed with training updates but it does not impact the overall performance. To verify this claim, we did experiments by updating CAVs in the student after every few training iterations (50, 100, 200, etc.). We observed no significant performance changes due to this. This may be because the student's activation space changes gradually relative to the distillation which converges very fast (in less than an epoch i.e 200-500 iterations).

**Datasets:**
Instead of opting for multiple concepts for a relatively well-defined problem in classification, we opted to focus on more complex concepts like illumination-invariance in an ill-defined problem like IID to gauge the utility of technique in a more real-world scenario. MNIST datasets have been chosen because of two reasons:
1) Synthetic biases in MNIST datasets are designed to be extreme and hence more challenging than a real world scenario. Introduction of abstract biases in a well-understood dataset helps gauge the impact of the technique in a better way.
2) MNIST allows comparisons with other contemporary works, which all report using the same methodology.
Apart from this we introduced a more challenging TextureMNIST dataset and also report results on BFFHQ which has natural images. In case of IID, IIW benchmark is the only large scale human annotated dataset (with rest of the datasets like Sintel being synthetically rendered with known issues in the IID evaluation).

We will be making the above more clear in the paper. Additionally, we will improvise the figures and pipeline description.

**Common Citations:**
* M. Xue, Q. Huang, H. Zhang, L. Cheng, J. Song, M. Wu, and M. Song. Protopformer: Concentrating on pro-totypical parts in vision transformers for interpretable image recognition.
* M. Keswani, S. Ramakrishnan, N. Reddy, and V. N. Balasubramanian. Proto2proto: Can you recognize the car, the way i do?
* Xian, Y., Lampert, C. H., Schiele, B., & Akata, Z. (2018). Zero-shot learning—a comprehensive evaluation of the good, the bad and the ugly.
* Wang, Y., Yao, Q., Kwok, J. T., & Ni, L. M. (2020). Generalizing from a few examples: A survey on few-shot learning.

(Note) Notations for rebuttal: S:LXX denotes supplementary Line XX while LXX denotes main paper lines XX.

---

### Decision · Program_Chairs · 2023-09-21

**Decision:**

Accept (poster)

**Comment:**

After discussion, three of four reviewers recommend acceptance. The remaining reviewer has not updated their score after extensive discussion with the authors, but appears satisfied with the responses. The AC see no grounds to overturn this consensus. Authors are strongly encouraged to include the additional ablations and clarifications from the discussions in the final version.